# Effects of Crude *β*-Glucosidases from *Issatchenkia terricola*, *Pichia kudriavzevii*, *Metschnikowia pulcherrima* on the Flavor Complexity and Characteristics of Wines

**DOI:** 10.3390/microorganisms8060953

**Published:** 2020-06-24

**Authors:** Wenxia Zhang, Xuanhan Zhuo, Lanlan Hu, Xiuyan Zhang

**Affiliations:** 1College of Food Science and Technology, Huazhong Agricultural University, Wuhan 430070, China; zwx775201996@126.com (W.Z.); Zhuoxuanhanzxh@163.com (X.Z.); Hulanlan@webmail.hzau.edu.cn (L.H.); 2Hubei International Scientific and Technological Cooperation Base of Traditional Fermented Foods, Wuhan 430070, China

**Keywords:** *β*-glucosidase, non-*Saccharomyces* yeast, wine, flavor

## Abstract

To investigate the effects of crude *β*-glucosidases from *Issatchenkia terricola* SLY-4 (SLY-4E), *Pichia kudriavzevii* F2-24 (F2-24E), and *Metschnikowia pulcherrima* HX-13 (HX-13E) on flavor complexity and characteristics of wines, grape juice was fermented by *Saccharomyces cerevisiae* with the addition of SLY-4E, F2-24E and HX-13E, respectively. The growth and sugar consumption kinetics of *S. cerevisiae*, the physicochemical characteristics, the volatile compounds, and the sensory dimensions of wines were analyzed. Results showed that adding SLY-4E, F2-24E, and HX-13E into must had no negative effect on the fermentation and physicochemical characteristics of wines, but increased the content of terpenes, esters, and fatty acids, while decreased the C_6_ compound content. Each wine had its typical volatile compound profiles. Adding SLY-4E or F2-24E into must could significantly improve the flavor complexity and characteristics of wines. These results would provide not only an approach to improve flavor complexity and characteristics of wines, but also references for application of *β*-glucosidases from other sources.

## 1. Introduction

Wine is popular with customers for its high nutrition value and healthy functions. In 2018, wine production was 292,300,000 L around the world and 9,300,000 L in China (OIV, 2019). With the improvement of living standards and health awareness of consumers, wine will be more and more popular. However, the flavor of wine is still lacking in complexity and characteristics, which will affect the competitiveness of wine in the fruit wine market.

Wine flavor mainly depends on the varietal aroma compounds, fermentative aroma compounds and aged aroma compounds [1]. The varietal aroma compounds often exist in the form of non-volatile glycosides with no flavor, but their glycosidic bonds can be hydrolyzed by enzymes from microorganisms and the volatile varietal aroma compounds are released during fermentation which would contribute to the complexity and characteristics of wine [2]. Fermentative aroma compounds are secondary metabolites produced by yeasts [3], and aging aroma compounds result from the transformation of aroma compounds during aging [4].

Co-fermentation of non-*Saccharomyces* yeast and *Saccharomyces cerevisiae* can improve the flavor complexity and characteristics of wines [5,6,7], because non-*Saccharomyces* yeasts are rich in various enzymes which can hydrolyze and release abundant aroma compounds. *Issatchenkia terricola* SLY-4, *Pichia kudriavzevii* F2-24, *Metschnikowia pulcherrima* HX-13 strains with high *β*-glucosidase activity were applied to co-fermentation of wines with *S. cerevisiae* through simultaneous inoculation fermentation and sequential inoculation fermentation by our research group. The results indicated sequential inoculation fermentation of non-*Saccharomyces* yeasts and *S. cerevisiae* could effectively improve the wine flavor [8]. However, the non-*Saccharomyces* yeast strain is easily inhibited by *S. cerevisiae* for their nutrient competition or other killer factors during co-fermentation [9,10,11], and will lead to the prolonged fermentation period or even the failure of co-fermentation. Our previous results showed that the fermentation periods of sequential inoculation were prolonged by five days, leading to an increase of production cost.

Addition of exogenous enzymes into must during wine fermentation can also improve flavor complexity and characteristics of wines [12], which can also ensure the success of fermentation process. Most enzymes used in oenology are from fungus, such as *Aspergillus niger* and *Trichoderma harzianum*, which are a mixture of non-specific enzymes and might trigger non-specific hydrolysis reactions or form undesirable flavors [13]. Among those enzyme preparations, *β*-glucosidase plays an important role in releasing varietal aroma compounds and increases the flavor complexity and characteristics of wines. Crude *β*-glucosidases were extracted from *I. terricola* SLY-4 (SLY-4E), *P. kudriavzevii* F2-24 (F2-24E), and *M. pulcherrima* HX-13 (HX-13E) in our previous studies, but it was still unknown whether they can be used to improve the flavor complexity and characteristics of wines or not.

Therefore, to explore the effects of SLY-4E, F2-24E, and HX-13E on flavor complexity and characteristics of wines, must was fermented by *S. cerevisiae* with addition of SLY-4E, F2-24E, and HX-13E, respectively. The growth and the sugar consumption kinetics of *S. cerevisiae*, the volatile compounds and the sensory of wines will be analyzed. The research results would provide not only an approach to improve the flavor complexity and characteristics of wines, but also references for the application of *β*-glucosidases from other sources.

## 2. Materials and Methods

### 2.1. Strains and Media

*I. terricola* SLY-4, *P. kudriavzevii* F2-24, and *M. pulcherrima* HX-13 with high *β*-glucosidase activity were kept in our laboratory. *S. cerevisiae* was a commercial strain purchased from Actiflore^®^ F33 (Laffort, Bordeaux, France).

Yeast extract peptone dextrose medium (YPD, 10 g/L yeast extract, 20 g/L peptone, 20 g/L glucose) was used for starter preparation. YPD solid medium was prepared by adding 20 g/L agar into YPD for *S. cerevisiae* cell counting.

Fermentation medium (10 g/L yeast extract, 20 g/L peptone, 20 g/L glucose, 3 g/L NH_4_NO_3_, 4 g/L KH_2_PO_4_, 0.5 g/L MgSO_4_·7H_2_O, 10 mL/L Tween 80) was used for culturing the non-*Saccharomyces* yeast strains to produce *β*-glucosidases.

### 2.2. Extraction of Crude β-glucosidase

The starter cultures of non-*Saccharomyces* yeast strains were inoculated into 40 mL fermentation medium contained in 250 mL flask and cultured at 28 °C at 120 rpm to reach the maximum enzyme activities. Each culture was centrifuged for 5 min (4 °C, 8500 rpm), and the supernatant was collected. The crude *β*-glucosidase was prepared through ammonium sulfate precipitation as following steps: the (NH_4_)_2_SO_4_ powder was added into the supernatant slowly till reaching 80% saturation on ice, then the solution was left overnight at 4 °C. Then the solution was centrifuged for 30 min (4 °C, 13,000 rpm), and the precipitate was collected. After that the 0.01 mol/L phosphate buffer (pH 5.0) was used to dissolve the precipitate, and then dialyzed in dialysis tubing (the nominal cut-off was 3500 Da, and was prepared in boiling water for 15 min) with the same buffer until no ammonium ions were present. The dialyzed enzyme solution was concentrated by using PEG 20000.

The *β*-glucosidase activities of non-*Saccharomyces* strains were monitored by using the following approach: 0.1 mL enzyme solution with 0.2 mL *p*-nitrophenyl *β*-D-glucopyranoside solution (PNPG in the phosphate buffer, 5 mmol/L) was incubated at 50 °C for 30 min, then the reaction was inactivated by adding 2.0 mL 1.0 mol/L Na_2_CO_3_ into reaction solution and the absorption value at 400 nm of solution was measured. One unit of activity (U) was defined as the amount of enzyme required to produce 1.0 μmol *p*-nitrophenol per minute under the above reaction condition.

### 2.3. Preparation of Wine

The preparation of wine was performed as described by Shi et al. [8]. Cabernet Sauvignon grapes were harvested from the Helan Mountain region vineyard (Ningxia province, China) in 2017. The grape must was prepared and analyzed (residual sugar 239.9 g/L, total acidity 7.1 g/L expressed as tartaric acid) after stemming and crushing. 800 mL grape must was added into 1 L glass bottle and pasteurized at 68 °C for 30 min. The grape must was macerated at 4 °C for 12 h after 50 mg/L SO_2_ was added, and then the *S. cerevisiae* was inoculated at 10^6^ CFU/mL and 1 U/L SLY-4E, F2-24E or HX-13E was added into the pasteurized must at the same time. After that the must was fermented at 20 °C by using *S. cerevisiae*. Pure culture of *S. cerevisiae* without enzyme addition was used as the control. Each experiment was conducted in triplicate.

### 2.4. Growth and Sugar Consumption Kinetics of Saccharomyces Cerevisiae

Samples were taken during fermentation on each day. Then the yeast cell numbers in samples were counted on YPD plate through dilution coating plate method and the residual sugar was determined through the method recommended by the International Organization of Vine and Wine (OIV, 2005).

### 2.5. Analysis of Physicochemical Characteristics and Volatile Compounds in Wines

Alcohol, total acidity, and volatile acidity were determined through methods recommended by the International Organization of Vine and Wine (OIV, 2005). Headspace solid-phase micro-extraction with 50/30 μm DVB/CAR/PDMS fiber (Supelco, Bellefonte, PA, USA) was used to extract the volatile compounds which were then analyzed with an Agilent 6890N gas chromatograph with a DB-5 capillary column (30 m × 0.32 mm × 0.25 μm) coupled to an Agilent 5975B mass spectrometer (GC-MS) according to Shi et al. [8] with little modification. A total of 8 mL wine, 2 g NaCl, and 450 μg/L cyclohexanone (internal standard) were added into a 20 mL headspace bottle and stirred with a magnetic bar in 40 °C water bath for 15 min. The fiber was pushed into the headspace of the sample for 30 min and immediately desorbed in the injector of gas chromatography at 250 °C for 5 min. GC analysis conditions were as following: temperature increased from 40 °C to 130 °C at 3 °C/min, and then to 250 °C at 4 °C/min. The injector and detector temperatures were set at 250 °C and 260 °C, respectively. The mass spectrometer was operated in electron impact ionization mode at 70 eV, and ion source temperature was 250 °C. Linalool, phenylethyl alcohol, isoamyl alcohol, octanoic acid, and ethyl hexanoate were identified by comparing their retention time and MS-spectra with those of the corresponding standard compounds, and other volatile compounds were identified by comparing the MS fragmentation pattern of each compound with that in database Wiley 7.0 and NIST05. The following formula was used for the calculation of the compound content: (1)$$\mathrm{Compound}\;\mathrm{content}\;(\mu g/\mathrm{mL})=\frac{\mathrm{GC}\;\mathrm{peak}\;\mathrm{areas}\;\mathrm{of}\;\mathrm{the}\;\mathrm{compound}\;\times \;\mathrm{Quality}\;\mathrm{of}\;\mathrm{internal}\;\mathrm{standard}\;(\mu g)}{\mathrm{GC}\;\mathrm{peak}\;\mathrm{area}\;\mathrm{of}\;\mathrm{the}\;\mathrm{internal}\;\mathrm{standard}\;\times \;\mathrm{Volume}\;\mathrm{of}\;\mathrm{the}\;\mathrm{sample}\;(\mathrm{mL})}$$

### 2.6. Sensory Evaluations of Wine

The sensory evaluation was performed as described by Belda et al. [14] with modifications. Wines were evaluated by ten well-trained panelists (five females and five males) in a tasting room at about 20 °C. Approximately 20 mL wine samples were poured into wine glasses and presented in triplicate. The panelists smelled the aroma for 5 to 10 s. Potable water was provided for rinsing the palate during testing. Sensory descriptions including floral, fruity, appearance, mouth feel and acceptance of wine were scored from zero (weak) to nine (intense), respectively.

### 2.7. Data Analyses

Microsoft Office 2016 and GraphPad Prism 6.0 were used to complete the data and charts. SPSS 19.0 software (SPSS Inc., Chicago, IL, USA) was exerted to do one-way analysis of variance (ANOVA) and multiple mean comparisons were completed by Duncan test. Principal component analysis (PCA) was used to reveal the correlation and segregation of varietal aroma compounds with different wines and to reveal the correlation among volatile components, sensory evaluation and wines by SIMCA-P 14.1 (Umetrics AB, Umea, Sweden). Hierarchical clustering and heat map visualization of fermentative aroma compounds in different wines were performed with MultiExperiment Viewer 4.9.0 (TIGR, Sacramento, CA, USA) after the Z-score standardization.

## 3. Results and Discussions

### 3.1. Yeast Growth and Sugar Consumption Kinetics during Wine Fermentation

The maximum biomass of *S. cerevisiae* in wines added with SLY-4E (SLY-4EW, 4.01 × 10^10^ CFU/mL), wines added with F2-24E (F2-24EW, 2.11 × 10^10^ CFU/mL) and wines added with HX-13E (HX-13EW, 9.73 × 10^9^ CFU/mL) on the 8th day were higher than that of Control (5.14 × 10^9^ CFU/mL), and there were significant differences in maximum biomass of *S. cerevisiae* among SLY-4EW, F2-24EW, and HX-13EW (Figure 1). The reason that the maximum biomass of *S. cerevisiae* in SLY-4EW, F2-24EW, and HX-13EW was higher than that of the control was still unclear.

The residual sugar contents in SLY-4EW, F2-24EW, and HX-13EW were 5.83 g/L, 5.03 g/L, and 5.57 g/L, respectively, which were significantly higher than that in the control (3.98 g/L) (Figure 1). The results indicated that adding crude *β*-glucosidases into must significantly increased the residual sugar content in wines. Moreover, there were significant differences in residual sugar content among SLY-4EW, F2-24EW, and HX-13EW, which might be caused by the glucose released from the aroma precursors hydrolyzed by different *β*-glucosidases.

### 3.2. The Physicochemical Characteristics and the Volatile Compounds of Wines

The analysis results indicated that all the physicochemical characteristics of wines met the standards of the International Organization of Vine and Wine for dry wine (data not shown). At the same time, the analysis results of volatile compounds indicated that 67 kinds of volatile compounds in wines were detected and were categorized into the varietal aroma compounds and the fermentative aroma compounds (Table 1).

#### 3.2.1. Varietal Aroma Compounds

A total of 17 kinds of varietal aroma compounds were quantified and classified into three kinds of C_6_ compounds, three kinds of terpenes, one kind of C_13_-norisoprenoid, and 10 kinds of benzene derivatives. Although compared to the control (19.4 mg/L), the total varietal aroma compound content showed a downward tendency in enzyme-treated wines (16.7 mg/L in F2-24EW and 12.5 mg/L in HX-13EW) except in SLY-4EW (29.0 mg/L), a different tendency of specific compound levels were observed. On the other hand, there was a significant difference in the content of varietal aroma compounds among SLY-4EW, F2-24EW, and HX-13EW, which suggested that the content of varietal aroma compounds may relate to the *β*-glycosidase activity from different strains.

C_6_ compounds are derived from long-chain fatty acids in grapes during berry ripening and crushing, which would take on an unpleasant grassy note to wines. In this study, the total C_6_ compound content presented a significantly decrease in SLY-4EW, F2-24EW, and HX-13EW. Previous study also showed that the *β*-glucosidase from *Rhodotorula mucilaginosa* could decrease C_6_ compound content in wines [25]. This indicated that the activity of the crude *β*-glucosidases significantly affected C_6_ compound levels, so SLY-4E, F2-24E, and HX-13E addition might effectively reduce the unpleasant green note of wines.

Terpenes and *β*-damascenone are derived from the precursors on grape hydrolyzed by glycosidase [26]. In wines, terpenes have positive contribution to the floral and fruity notes and *β*-damascenone are responsible for both floral and sweet flavor [20]. Table 1 showed that higher terpene and *β*-damascenone contents were detected in SLY-4EW, F2-24EW, and HX-13EW, except the *β*-damascenone was not detected in HX-13EW. Sun et al. [25] also reported that *S. cerevisiae* fermentation simultaneously added with *β*-glycosidase extracted from *Hanseniaspora uvarum* increased terpene and C_13_-norisoprenoid levels in wines. Additionally, the one-way ANOVA results indicated that there were significant differences of terpene and *β*-damascenone contents among wines treated by different *β*-glucosidase, and this observation could be explained by different hydrolyzed ability of glycosidases towards different substrates. These results suggested that adding crude *β*-glucosidase into must could improve the concentration of terpene and *β*-damascenone, thus providing the fruity and floral flavor to wines.

Compared with that in the Control, higher content of benzene derivatives was detected in SLY-4EW, which was also reported by Hu et al. [27] that the benzene derivative contents in wines were significantly increased after adding *β*-glucosidases from *H. uvarum* and *A. niger* into must, respectively. However, the reason that leading to the lower contents of benzene derivatives in F2-24EW and HX-13EW remained to be investigated. On the other hand, the one-way ANOVA results revealed that adding SLY-4E, F2-24E and HX-13E into must had significantly different effects on the benzene derivative levels, which might cause different flavor profiles through interactions between compounds.

A principal component analysis (PCA) was carried out to reveal the correlation and segregation of varietal aroma compounds with different wines (Figure 2). Here 77.5% variance was explained by 17 different varietal aroma compounds, and PC1 and PC2 accounted for 52.7% and 24.8% of the total variance, respectively. The distribution of wine samples implied that addition of SLY-4E had a positive effect on various varietal compounds, such as citronellol, geraniol, and *β*-damascenone, and F2-24E was related to the compounds like linalool and phenyl octanoate, while HX-13EW was located at the lower right quadrant with (E)-3-hexen-1-ol and the Control was located at the upper right quadrant with benzaldehyde and 1-hexanol. These results illustrated that the SLY-4E and F2-24E could be added into must to produce wines with more kinds of varietal aroma compounds with fruity and floral note, whereas the HX-13E addition would present a little bit green note to wines. Additionally, there was a significant difference in varietal aroma compounds profile among the enzyme-treated wines, which would take on typical flavor characteristics in wines.

#### 3.2.2. Fermentative Aroma Compounds

Fifty fermentative aroma compounds were identified, including six higher alcohols, 32 esters, seven fatty acids, and five kinds of carbonyl compounds. The total contents of fermentation compounds in enzyme-treated wines were significantly improved compared to that of the control. The higher content of fermentative compounds suggested that glycosidase activities contributed to their increase, especially for SLY-4E which had the highest concentration (80.8 mg/L), followed by F2-24E (63.0 mg/L) and HX-13E (61.0 mg/L).

Higher alcohols have considerable positive contribution to aromatic complexity of wines with the concentration below 300 mg/L [28], and all wine samples detected in this study met the concentration standard. F2-24EW and HX-13EW had lower concentration of higher alcohols than the control, whereas the SLY-4EW had higher content of higher alcohol, which was consistent with a previous study showed the addition of *β*-glucosidase from *H. uvarum* significantly increased the content of higher alcohols [29]. Furthermore, the results of one-way ANOVA in Table 1 suggested that the content of higher alcohols were significantly influenced by adding different *β*-glucosidases into must. The significant difference of higher alcohol content among wine samples indicated that *β*-glucosidases from different strains differentially affected the formation of higher alcohols.

Excessive fatty acids in wines have been described with cheese, fatty and rancid notes, while fatty acids are also essential substrates for the synthesis of fatty acid esters, and their branched forms can suppress the animal note of ethylphenols [30,31]. In this study the content of fatty acids showed an upward tendency in all enzyme-treated wines, Hu et al. [27] also reported that the *β*-glucosidase from *R. mucilaginosa* significantly increased the fatty acid content, and the one-way ANOVA result indicated that the concentration of fatty acids was significantly affected by crude *β*-glucosidases from different strains, which indicated that addition of SLY-4E, F2-24E, and HX-13E into must might finally improve the flavor complexity of wines to different degrees.

Esters are known as the major contributors to the floral and fruity trait of wines [16], and they generally can be categorized into fatty acid ethyl esters, acetates and other esters. In this study, adding SLY-4E, F2-24E, and HX-13E into must brought a dramatically increase in total ester (including the fatty acid ethyl esters, acetates and other esters) levels compared to those of the control. Ma et al. [32] also reported that adding crude enzymes (including esterase and *β*-glucosidase) from *Pichia fermentans* into must significantly increased the contents of esters. As a consequence, adding SLY-4E, F2-24E, and HX-13E into must might contribute to the floral and fruity flavor in the wines for the higher content of esters.

The detected carbonyl compounds in wines might have negative effects on the flavor of wines (showed in Table 1), but the real effects of these compounds on wines should be further analyzed. From the analyses of the one-way ANOVA a conclusion can be pointed that there was a significant difference of the carbonyl compound content among SLY-4EW, F2-24EW, and HX-13EW. These results indicated that the *β*-glucosidases from different strains had different effects on the content of carbonyl compounds in wines, but the specific effects of SLY-4E, F2-24E, and HX-13E on wines should be further studied combined to sensory evaluation method.

To visualize the differences of fermentative aroma compounds in different wines, the hierarchical cluster analysis of these compounds was conducted (Figure 3). The results showed that wines were classified into three groups including SLY-4EW/HX-13EW, F2-24EW, and control. All the detected fermentative aroma compounds were clustered into four classes and designated as class I, II, III, and IV. Furthermore, compared with the control, which was rich in class I (ethyl heptanoate, isobutyric acid, decanal, decanoic acid) and some kinds of compounds in class II (butyrolactone, 2,3-butanediol, propyl decanoate, isoamyl alcohol, nonanal), F2-24EW was abundant in class IV (pentyl acetate, methyl caprylate, caproic acid, ethyl nonanoate, isovaleric acid, 2,3-pentanedione, 2-methylcyclopentanol, ethyl caprylate) and some kinds of class III compounds (ethyl hexanoate, ethyl caprate, octyl acetate, isopropyl myristate, 3-methyl-1-pentanol, 2-methylbutyric acid, 1-octanol, ethyl laurate), and methyl hexanoate belonged to class I. HX-13EW had higher contents of compounds from class II (isoamyl acetate, ethyl butanoate, ethyl propionate, octanoic acid) and class III (ethyl laurate, ethylhexadec-9-enoate, isoamyl hexanoate, butanoic acid, isoamyl caprylate, 1-octanol, 2-methylbutyric acid, ethyl hexanoate, ethyl caprate). SLY-4EW was abundant in compounds belonging to class II (3-methyl-1-pentanol, isoamyl alcohol, nonanal, geranylacetone, isoamyl dodecanoate, isoamyl laurate, 1-decanol, ethyl butanoate, ethyl propionate, octanoic acid), class III (1-octanol, 2-methylbutyric acid, ethyl hexanoate, ethyl caprate), class IV (ethyl nonanoate, isovaleric acid, 2,3-pentanedione, ethyl caprylate), and ethyl heptanoate, belonging to class I. The observation suggested that adding SLY-4E, F2-24E, and HX-13E into must significantly increased the kind and content of fermentation aroma compounds in wines, which would contribute to the flavor complexity of wines. Also adding different enzymes into must produced different profiles of fermentative compounds which would impart different flavor complexities on wines.

### 3.3. Sensory Evaluation of Wine Samples

The results of the sensory evaluation of wines (Figure 4) showed SLY-4EW and F2-24EW obtained insignificantly different scores in floral (8.15 and 8.61), fruity (8.21 and 8.74), appearance (7.75 and 7.59), mouth feel (7.81 and 8.26), and acceptance (8.55 and 8.60), which were higher than those of the control. On the other hand, the scores of floral (7.18), fruity (7.05) and acceptance (7.09) in HX-13EW had no significant difference from those in the control. The results indicated that addition of SLY-4E or F2-24E into must was a better choice to produce wines with outstanding floral, fruity, appearance, mouth feel, and acceptance.

In order to reveal the correlation among volatile components, sensory evaluation, and wines treated by crude *β*-glucosidases from different non-*Saccharomyces* yeasts, the PCA was conducted (Figure 5). The results suggested that the main indicators of sensory evaluation were more related to the esters, C_13_-norisoprenoids, and aldehydes and ketones. On the other hand, addition of SLY-4E or F2-24E into must contributed positively to the sensory quality of wines, which was also reflected in the sensory evaluation results. However, the HX-13EW along with the control was more affected by the C_6_ compounds which would impart an unpleasant grassy note and have a negative effect on the sensory evaluations of wines. Those results indicated that the addition of SLY-4E or F2-24E into must is greatly potential to produce wines with better sensory evaluation.

## 4. Conclusions

Adding SLY-4E, F2-24E, and HX-13E into must had no negative effect on the growth of *S. cerevisiae* and the physicochemical characteristics of wines, but they could reduce the C_6_ compound content and increase the contents of terpenes and *β*-damascenone (except for HX-13E). At the same time, adding crude *β*-glucosidases into must could also increase the kind and content of fermentative aroma compounds, especially the esters and fatty acids, which would reduce the unpleasant green note and enhance the fruity, floral, and sweet flavor. Furthermore, adding different crude *β*-glucosidases from different yeast strains into must produced different typical varietal aroma compounds and different fermentative aroma compound profiles which could present on typical flavor characteristics in wines. More importantly, adding SLY-4E and F2-24E into must significantly improved the flavor complexity and characteristics of wines, although adding HX-13E had no significant effect on them. Therefore, SLY-4E and F2-24E could be used to improve the flavor complexity and characteristics of wines. Those results would provide not only an approach to improve flavor complexity and characteristics of wine, but also references for the application of *β*-glucosidases from other sources.

## Figures and Tables

**Figure 1 microorganisms-08-00953-f001:**
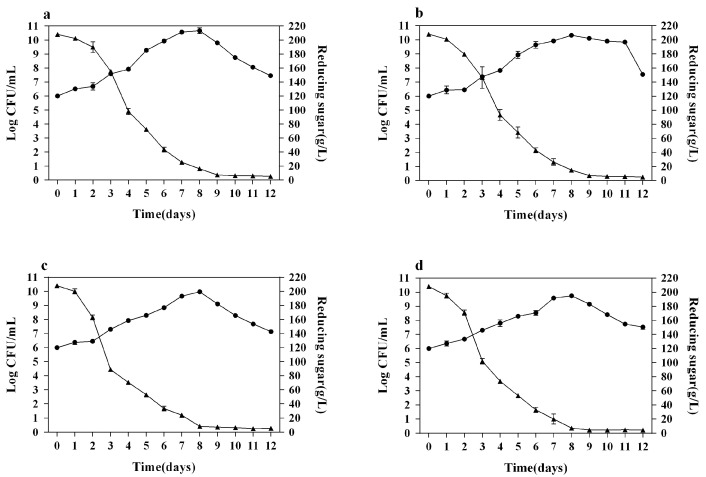
Growth kinetics and sugar consumption kinetics of *S. cerevisiae* during wine fermentation. (**a**) Wines treated by crude *β*-glucosidase from *I. terricola* SLY-4; (**b**) Wines treated by crude *β*-glucosidase from *P. kudriavzevii* F2-24; (**c**) Wines treated by crude *β*-glucosidase from *M. pulcherrima* HX-13; (**d**) Wines without enzyme treatment. -●- growth kinetics of *S. cerevisiae*; -▲- sugar consumption kinetics.

**Figure 2 microorganisms-08-00953-f002:**
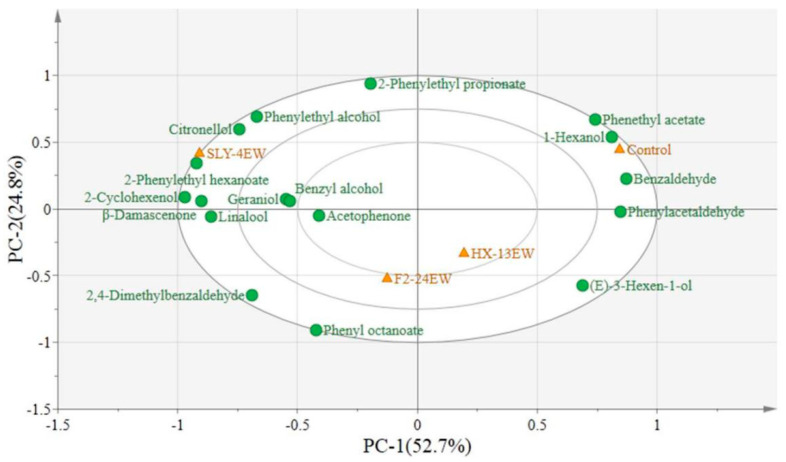
Bioplot of PCA for varietal aroma compounds from wines treated by crude *β*-glucosidase from different non-*Saccharomyces* yeasts. SLY-4EW: Wines treated by crude *β*-glucosidase from *I. terricola* SLY-4; F2-24EW: Wines treated by crude *β*-glucosidase from *P. kudriavzevii* F2-24; HX-13EW: Wines treated by *β*-glucosidase from *M. pulcherrima* HX-13; Control: Wines without enzyme treatment. 
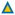
: Wine samples; 
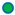
: Varietal aroma compounds.

**Figure 3 microorganisms-08-00953-f003:**
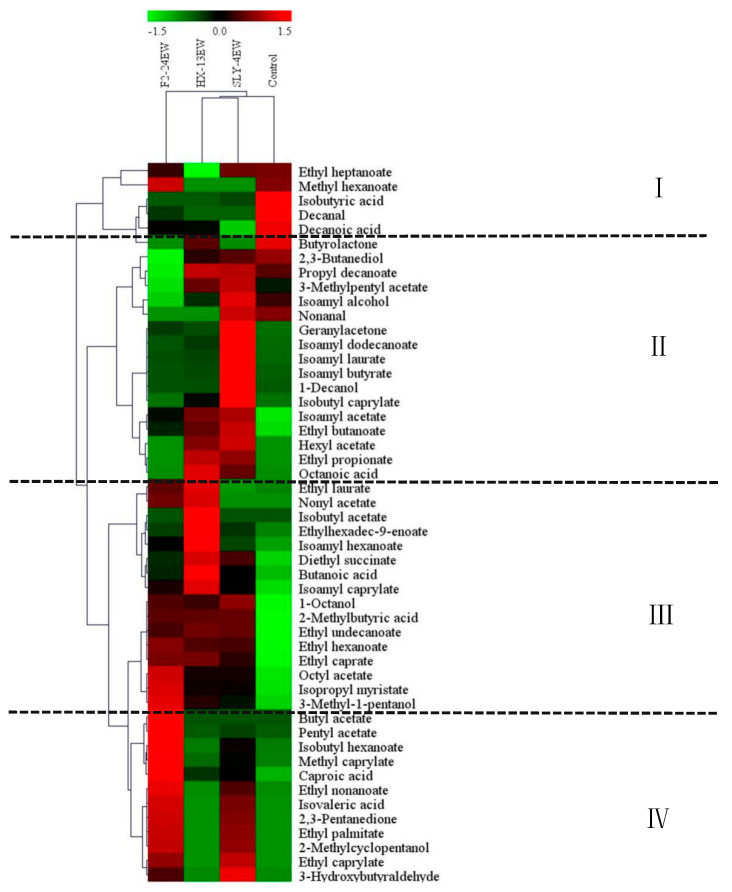
Hierarchical clustering and heat map visualization of fermentative aroma compounds from wines treated by crude *β*-glucosidase from different non-*Saccharomyces* yeasts. SLY-4EW: Wines treated by crude *β*-glucosidase from *I. terricola* SLY-4; F2-24EW: Wines treated by crude *β*-glucosidase from *P. kudriavzevii* F2-24; HX-13EW: Wines treated by *β*-glucosidase from *M. pulcherrima* HX-13; Control: Wines without enzyme treatment. All the detected fermentative aroma compounds were clustered into four classes and designated as class I, II, III, and IV.

**Figure 4 microorganisms-08-00953-f004:**
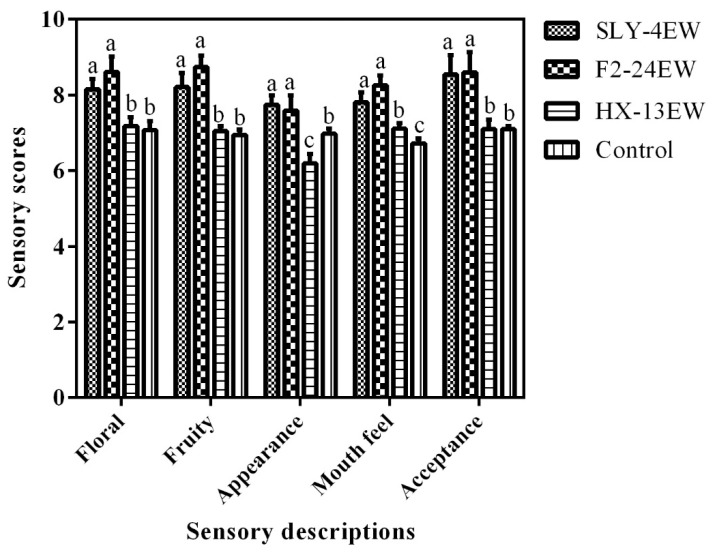
Sensory evaluation of wines treated by crude *β*-glucosidase from different non-*Saccharomyces* yeasts. Different letters within a cluster indicated differences among wine samples determined by the Duncan test at a 95% confidence level. SLY-4EW: Wines treated by crude *β*-glucosidase from *I. terricola* SLY-4; F2-24EW: Wines treated by crude *β*-glucosidase from *P. kudriavzevii* F2-24; HX-13EW: Wines treated by *β*-glucosidase from *M. pulcherrima* HX-13; Control: Wines without enzyme treatment.

**Figure 5 microorganisms-08-00953-f005:**
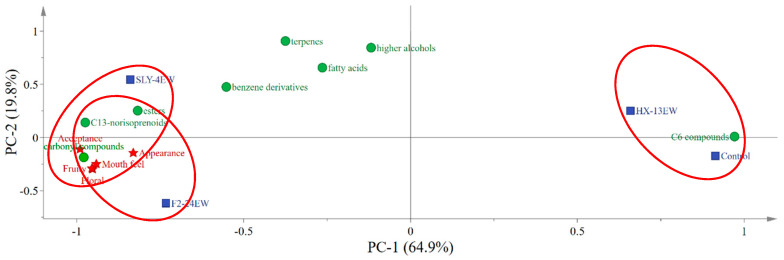
Bioplot of PCA of volatile components, sensory evaluation and wines treated by crude *β*-glucosidase from different non-*Saccharomyces* yeasts. SLY-4EW: Wines treated by crude *β*-glucosidase from *I. terricola* SLY-4; F2-24EW: Wines treated by crude β-glucosidase from *P. kudriavzevii* F2-24; HX-13EW: Wines treated by *β*-glucosidase from *M. pulcherrima* HX-13; Control: Wines without enzyme treatment. 
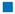
: Wine samples; 
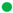
: Volatile compounds; 
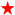
: Sensory description. The volatile components, sensory evaluation and wine samples in one red circle represented the closer correlation.

**Table 1 microorganisms-08-00953-t001:** Contents of volatile compounds detected in sample wines by GC-MS.

Compounds	Concentration (mg/L)				Odor Threshold (mg/L)	Sensory Description
	SLY-4 EW	F2-24 EW	HX-13 EW	Control		
1-Hexanol	0.099 ± 0.010 ^c^	0.079 ± 0.016 ^d^	0.134 ± 0.005 ^b^	0.195 ± 0.008 ^a^	8 [15]	Herbaceous, grass [15]
2-Cyclohexenol	0.042 ± 0.005 ^a^	0.034 ± 0.002 ^b^	0.026 ± 0.002 ^c^	0.023 ± 0.001 ^c^	-	-
(*E*)-3-Hexen-1-ol	0.023 ± 0.005 ^c^	0.045 ± 0.001 ^b^	0.064 ± 0.005 ^a^	0.047 ± 0.009 ^b^	0.4 [16]	Herbaceous, green [15]
C_6_ compounds	0.164 ± 0.02 ^c^	0.158 ± 0.019 ^c^	0.224 ± 0.012 ^b^	0.265 ± 0.018 ^a^		
Linalool	0.040 ± 0.011 ^a^	0.035 ± 0.007 ^a^	Nd	Nd	0.025 [16]	Muscat, flowery, fruity [15]
Citronellol	0.143 ± 0.018 ^a^	0.058 ± 0.005 ^c^	0.081 ± 0.002 ^b^	0.072 ± 0.006 ^b,c^	0.01 [17]	Green lemon [15]
Geraniol	0.146 ± 0.007 ^a^	Nd	0.152 ± 0.022 ^a^	Nd	0.03 [18]	Roses, geranium [18]
Terpenes	0.329 ± 0.036 ^a^	0.093 ± 0.012 ^c^	0.233 ± 0.024 ^b^	0.072 ± 0.006 ^c^		
*β*-Damascenone	0.037 ± 0.000 ^a^	0.026 ± 0.001 ^b^	Nd	Nd	5*10^−5^ [15]	Floral, sweet, apple [15]
C_13_-Norisoprenoids	0.037 ± 0.000 ^a^	0.026 ± 0.001 ^b^	Nd	Nd		
Benzaldehyde	Nd	Nd	0.019 ± 0.003 ^b^	0.024 ± 0.002 ^a^	2 [15]	Bitter almond, nut [15]
Benzyl alcohol	0.038 ± 0.004 ^a^	Nd	0.041 ± 0.002 ^a^	Nd	200 [17]	Almond, fatty [17]
Phenylacetaldehyde	0.143 ± 0.005 ^a^	0.171 ± 0.026 ^a^	0.155 ± 0.045 ^a^	0.184 ± 0.034 ^a^	0.001 [16]	Rose, floral, chocolate [16]
Acetophenone	0.005 ± 0.000 ^b^	Nd	0.007 ± 0.001 ^a^	Nd	-	-
Phenylethyl alcohol	26.7 ± 1.84 ^a^	14.9 ± 1.9 ^b^	10.3 ± 1.90 ^c^	16.6 ± 1.00 ^b^	14 [19]	Violet, rose, jasmine [19]
2,4-Dimethylbenzaldehyde	0.145 ± 0.021 ^a^	0.152 ± 0.013 ^a^	0.170 ± 0.018 ^a^	Nd	-	-
Phenethyl acetate	1.30 ± 0.192 ^b,c^	1.08 ± 0.156 ^c^	1.35 ± 0.102 ^b^	2.22 ± 0.000 ^a^	0.25 [16]	Flowery, pollen, perfume [16]
2-Phenylethyl propionate	0.027 ± 0.001 ^a^	0.010 ± 0.001 ^b^	0.017 ± 0.006 ^b^	0.023 ± 0.007 ^a,b^	-	-
2-Phenethyl hexanoate	0.091 ± 0.006 ^a^	0.030 ± 0.011 ^b^	Nd	Nd	-	-
Phenyl octanoate	0.012 ± 0.003 ^b^	0.026 ± 0.002 ^a^	0.020 ± 0.003 ^c^	Nd	-	-
Benzene derivative	28.45 ± 2.07 ^a^	16.41 ± 2.10 ^b^	12.1 ± 2.08 ^c^	19.1 ± 1.05 ^b^		
Varietal aroma	29.0 ± 2.13 ^a^	16.7 ± 2.14 ^b^	12.6 ± 2.12 ^c^	19.4 ± 1.07 ^b^		
Isoamyl alcohol	35.2 ± 0.063 ^a^	23.5 ± 1.49 ^d^	28.0 ± 2.27 ^c^	31.0 ± 1.41 ^b^	30 [15]	Whiskey, malt, burnt [15]
2,3-Butanediol	0.379 ± 0.034 ^a^	Nd	0.336 ± 0.035 ^a,b^	0.438 ± 0.017 ^a^	120 [19]	Butter, creamy [19]
1-Octanol	0.160 ± 0.011 ^a^	0.137 ± 0.034 ^a^	0.130 ± 0.009 ^a^	Nd	0.9 [19]	Flesh orange, rose, sweet herb [19]
1-Decanol	1.37 ± 0.199 ^a^	0.026 ± 0.002 ^b^	0.039 ± 0.001 ^b^	Nd	0.4 [16]	Orange flowery, special fatty [16]
2-Methylcyclopentanol	0.231 ± 0.008 ^a^	0.266 ± 0.108 ^a^	Nd	Nd	-	-
3-Methyl-1-pentanol	0.259 ± 0.090 ^b^	0.569 ± 0.002 ^a^	0.346 ± 0.041 ^b^	Nd	0.5 [19]	Soil, mushroom [20]
Higher alcohols	37.6 ± 0.405 ^a^	24.5 ± 1.63 ^c^	28.9 ± 2.36 ^b^	31.4 ± 1.43 ^b^		
Butanoic acid	0.034 ± 0.016 ^a^	0.026 ± 0.003 ^a^	0.073 ± 0.089 ^a^	Nd	0.173 [21]	Rancid, cheese, sweat [21]
Isobutyric acid	0.006 ± 0.002 ^b^	Nd	Nd	0.104 ± 0.006 ^a^	2.3 [20]	Rancid, butter, cheese [19]
Isovaleric acid	0.322 ± 0.029 ^b^	0.426 ± 0.029 ^a^	Nd	Nd	0.033 [16]	Sweet, acid, rancid [16]
2-Methylbutyric acid	0.081 ± 0.000 ^a^	0.079 ± 0.004 ^a^	0.080 ± 0.021 ^a^	Nd	0.033 [19]	Cheese [19]
Caproic acid	0.315 ± 0.016 ^b^	0.498 ± 0.043 ^a^	0.268 ± 0.088 ^b^	0.166 ± 0.020 ^c^	0.42 [22]	Cheese, rancid [22]
Octanoic acid	1.06 ± 0.065 ^b^	Nd	1.55 ± 0.105 ^a^	Nd	0.5 [19]	Rancid, harsh, cheese, fatty acid [19]
Decanoic acid	0.154 ± 0.045 ^b^	0.294 ± 0.129 ^a,b^	0.279 ± 0.029 ^a,b^	0.427 ± 0.134 ^a^	1 [19]	Fatty, unpleasant [19]
Fatty acids	1.97 ± 0.173 ^a^	1.32 ± 0.208 ^b^	2.25 ± 0.332 ^a^	0.697 ± 0.16 ^c^		
Ethyl propionate	0.367 ± 0.006 ^b^	Nd	0.420 ± 0.010 ^a^	Nd	1.8 [15]	Banana, apple [15]
Ethyl butanoate	0.528 ± 0.020 ^a^	0.244 ± 0.055 ^c^	0.411 ± 0.037 ^b^	Nd	0.02 [15]	Sour fruit, strawberry [15]
Ethyl hexanoate	5.77 ± 0.294 ^a^	6.66 ± 1.662 ^a^	5.87 ± 1.150^2 a^	Nd	0.005 [15]	Fruity, green apple, floral, violet [15]
Ethyl heptanoate	0.066 ± 0.015 ^a^	0.057 ± 0.003 ^a^	Nd	0.067 ± 0.022 ^a^	0.22 [15]	Pineapple [15]
Diethyl succinate	0.839 ± 0.074 ^b^	0.522 ± 0.054 ^c^	1.19 ± 0.267 ^a^	0.046 ± 0.012 ^d^	200 [19]	Light fruity [19]
Ethyl caprylate	12.7 ± 0.501 ^a^	11.1 ± 0.906 ^b^	Nd	Nd	0.005 [22]	Fruity, pineapple, pear, floral [22]
Ethyl nonanoate	0.158 ± 0.038 ^b^	0.257 ± 0.025 ^a^	Nd	Nd	1.3 [19]	Waxy, fruity [19]
Ethylhexadec-9-enoate	0.268 ± 0.066 ^b^	0.226 ± 0.017 ^b^	1.281 ± 0.336 ^a^	Nd	0.1 [19]	Green, fruity, fatty [19]
Ethyl caprate	7.23 ± 1.41 ^a^	8.78 ± 1.51 ^a^	8.65 ± 1.79 ^a^	Nd	0.5 [19]	Fruity, grape [19]
Ethyl undecanoate	0.030 ± 0.006 ^a^	0.028 ± 0.003 ^a^	0.031 ± 0.002 ^a^	Nd	-	-
Ethyl laurate	Nd	2.10 ± 0.38 ^b^	3.13 ± 0.819 ^a^	0.180 ± 0.041 ^c^	1.5 [16]	Flowery, fruity [16]
Ethyl palmitate	0.037 ± 0.006 ^a^	0.044 ± 0.008 ^a^	Nd	Nd	1 [15]	Wax, fatty [15]
Fatty acid ethyl esters	28.0 ± 2.43 ^a^	30.0 ± 4.62 ^a^	21.0 ± 4.42 ^b^	0.293 ± 0.075 ^c^		
Isoamyl acetate	7.22 ± 1.65 ^a^	4.07 ± 0.489 ^b^	6.32 ± 0.815 ^a^	Nd	0.03 [15]	Fresh, banana [15]
3-Methylpentyl acetate	0.372 ± 0.042 ^a^	0.042 ± 0.003 ^d^	0.308 ± 0.017 ^b^	0.205 ± 0.017 ^c^	-	-
Isobutyl acetate	Nd	Nd	0.061 ± 0.016 ^a^	Nd	1.6 [15]	Strawberry, fruity, flowery [19]
Butyl acetate	Nd	0.036 ± 0.007 ^a^	Nd	Nd	1.8 [23]	Pear, banana [23]
Pentyl acetate	0.014 ± 0.003 ^b^	0.217 ± 0.002 ^a^	Nd	Nd	-	-
Hexyl acetate	0.060 ± 0.005 ^a^	Nd	0.048 ± 0.008 ^b^	Nd	1.5 [19]	Pleasant fruity, pear [19]
Octyl acetate	0.072 ± 0.004 ^a^	0.081 ± 0.005 ^a^	0.072 ± 0.010 ^a^	0.057 ± 0.026 ^a^	0.012 [15]	Fruity, fennel, sweet [15]
Nonyl acetate	Nd	0.021 ± 0.003 ^a^	0.029 ± 0.008 ^a^	Nd	-	-
Acetic esters	7.74 ± 1.70 ^a^	4.47 ± 0.509 ^b^	6.84 ± 0.874 ^a^	0.262 ± 0.043 ^c^		
Butyrolactone	Nd	Nd	0.092 ± 0.015 ^b^	0.141 ± 0.003 ^a^	-	-
Isoamyl butyrate	0.080 ± 0.024 ^a^	0.003 ± 0.000 ^b^	0.005 ± 0.001 ^b^	Nd	-	-
Methyl hexanoate	Nd	0.005 ± 0.001 ^a^	Nd	0.004 ± 0.000 ^b^	-	-
Isobutyl hexanoate	0.004 ± 0.001 ^b^	0.010 ± 0.004 ^a^	Nd	Nd	-	-
Isoamyl hexanoate	0.175 ± 0.005 ^a^	0.182 ± 0.04 ^a^	0.207 ± 0.019 ^a^	0.166 ± 0.004 ^a,b^	-	-
Isobutyl caprylate	0.127 ± 0.005 ^a^	Nd	0.035 ± 0.009 ^b^	Nd	-	-
Isoamyl caprylate	0.323 ± 0.043 ^b^	0.377 ± 0.002 ^b^	0.625 ± 0.140 ^a^	Nd	-	-
Propyl decanoate	0.011 ± 0.000 ^a^	0.006 ± 0.000 ^c^	0.009 ± 0.001 ^b^	0.010 ± 0.001 ^a^	-	-
Isoamyl dodecanoate	3.03 ± 0.255 ^a^	0.163 ± 0.058 ^b^	0.392 ± 0.103 ^b^	Nd	-	-
Isoamyl laurate	0.168 ± 0.023 ^a^	0.012 ± 0.001 ^b^	0.016 ± 0.000 ^b^	Nd	-	-
Methyl caprylate	0.025 ± 0.005 ^b^	0.066 ± 0.017 ^a^	0.008 ± 0.001 ^c^	0.005 ± 0.000 ^c^	0.2 [24]	Intense citrus [24]
Isopropyl myristate	0.618 ± 0.035 ^b^	0.821 ± 0.049 ^a^	0.621 ± 0.066 ^b^	0.330 ± 0.022 ^c^	-	-
Other esters	4.56 ± 0.399 ^a^	1.65 ± 0.172 ^b^	2.01 ± 0.355 ^b^	0.656 ± 0.030 ^c^		
Esters	40.3 ± 4.54 ^a^	36.2 ± 5.30 ^b^	29.8 ± 5.64 ^b^	1.21 ± 0.148 ^c^		
Nonanal	0.021 ± 0.002 ^a^	Nd	Nd	0.017 ± 0.002 ^b^	0.015 [15]	Rose, almond [15]
Geranylacetone	0.046 ± 0.006 ^a^	0.028 ± 0.002 ^b^	0.027 ± 0.006 ^b^	0.025 ± 0.003 ^b^	-	-
Decanal	0.029 ± 0.008 ^b^	0.031 ± 0.004 ^a,b^	0.029 ± 0.014 ^b^	0.046 ± 0.004 ^a^	0.001 [16]	Green, fresh [16]
2,3-Pentanedione	0.741 ± 0.038 ^b^	0.929 ± 0.049 ^a^	Nd	Nd	<0.1 [15]	Butter, cheese [15]
3-Hydroxybutyraldehyde	0.062 ± 0.019 ^a^	0.037 ± 0.016 ^b^	Nd	Nd	-	-
Carbonyl compounds	0.899 ± 0.073 ^b^	1.03 ± 0.071 ^a^	0.056 ± 0.020 ^c^	0.088 ± 0.009 ^c^		
Fermentative aroma	80.8 ± 5.19 ^a^	63.0 ± 7.21 ^b^	61.0 ± 8.35 ^b^	33.4 ± 1.74 ^c^		

Data showed average of triplicates ± SD. Different letters within rows indicated differences among wine samples determined by the Duncan test at 95% confidence level, and “a” means the relatively highest content level, following by “b”, “c”, “d”; “Nd” means the compound was not detected by GC-MS in the corresponding wine sample; “-” means the odor threshold or the sensory description of the compound has not been reported in a literature. SLY-4EW: Wines treated by *I. terricola* SLY-4 crude *enzyme*; F2-24EW: Wines treated by *P. kudriavzevii* F2-24 crude enzyme; HX-13EW: Wines treated by *M. pulcherrima* HX-13 crude enzyme; Control: Wines without enzyme treatment.

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
