# Peer review of "Effects of Crude β-Glucosidases from Issatchenkia terricola, Pichia kudriavzevii, Metschnikowia pulcherrima on the Flavor Complexity and Characteristics of Wines"

_microorganisms, 2020, doi:10.3390/microorganisms8060953_

Round 1

Reviewer 1 Report

MICROORGANISMS_824381

Paper: “Effects of crude β-glucosidases from Issatchenkia terricola, Pichia kudriavzevii, Metschnikowia pulcherrima on the flavor complexity and characteristics of wines

Reviewer 1

  • The line numbers starts on page 9. Why?
  • Paragraph 2.5: The authors describe the GC-MS method used and and indicate 43°C as the temperature rate. Is this correct? Please check!
  • Table 1 is confusing: molecules are grouped by chemical family. I suggest to delete the bold.
  • In my opinion the discussion on volatile molecules needs to be improved

Author Response

Response to Reviewer 1 Comments

Thanks for your work of processing the manuscript (No: microorganisms-824381). Following are our response about your comments, and the revised parts within the manuscript have been in a red font.

Point 1: The line numbers starts on page 9. Why?

Response 1: The line numbers were renumbered during the process of changing the original manuscript into the specific version of the journal, and now we have renumbered them into the right version.

Point 2: Paragraph 2.5: The authors describe the GC-MS method used and and indicate 43°C as the temperature rate. Is this correct? Please check!

Response 2: The temperature rate was changed to the right number “4”, we are sorry for the mistake and thank you for your kind remind.

Point 3: Table 1 is confusing: molecules are grouped by chemical family. I suggest to delete the bold. 

Response 3: Thank you for your suggestion, we want to explained the reasons here. Since the aroma compounds in different chemical families have different odor descriptions, we grouped the molecules by chemical family in order to discuss the aroma compounds in an organized way and avoid to analyse more than 60 compounds one by one.

Point 4: In my opinion the discussion on volatile molecules needs to be improved

Response 4: The discussion on volatile molecules part have been improved, the revision parts within the manuscript were in red font.

Reviewer 2 Report

My main issue for this article is the quality of the volatile analysis. If I understood well the volatile compounds were semi-quantified. As the main focus of this paper is on the aroma analysis the authors should perform further analysis or rewrite the results and discussion part. Since compounds were semi-quantified the authors can't compare these values with those reported in literature or compare the concentrations found with the odor thresholds of  each compound.

Author Response

Response to Reviewer 2 Comments

Thanks for your work of processing the manuscript (No: microorganisms-824381). Following are our response about your comments, and the revised parts within the manuscript have been in a red font.

Point 1:My main issue for this article is the quality of the volatile analysis. If I understood well the volatile compounds were semi-quantified. As the main focus of this paper is on the aroma analysis the authors should perform further analysis or rewrite the results and discussion part. Since compounds were semi-quantified the authors can't compare these values with those reported in literature or compare the concentrations found with the odor thresholds of each compound.

Response 1: Thank you for your constructive suggestion and we have rewritten the results and discussion part and provided details about semi-quantitative analysis in Paragraph 2.5. Since what we needed to investigate was the relative decrease or increase of the compound levels due to the addition of the crude β-glucosidases instead of the specific values, and there were more than 60 kinds of volatile compounds to be detected in our study, we had to use the semi-quantitative analysis rather than the full-quantitative analysis to simplify the work. Considering your advice, the expression in the results and discussion part have changed, relative comparisons of compound contents were conducted, and we did not compare these specific values with those reported in literature, we just referred to literature in order to present the effects of β-glucosidases on the contents of compounds by comparing their contents in enzyme-treated wine samples to those in wine samples without enzyme-treated. On the other hand, the odor thresholds are just reference values, not fixed ones, and detecting odor thresholds of the 67 kinds of volatile compounds was a heavy workload, so we chose to obtain the odor thresholds of compounds from literature marked in Table 1, and compare the relative levels of compound contents in wine samples with thresholds reported in literature. Additionally, the discussion on volatile molecules part have been improved, the revision parts within the manuscript were in red font.

Round 2

Reviewer 1 Report

I really appreciate the review work and I think that the paper  can be accepted.

Reviewer 2 Report

I am satisfied and therefore I recommend acceptance in its current form